# Increasing turbidity in the North Sea during the 20[th] century due to changing wave climate

Robert J. Wilson[1,2], Michael R. Heath[2]

[1]Plymouth Marine Laboratory, Prospect Place, The Hoe, Plymouth, PL1 3DH, UK

[2]Department of Mathematics and Statistics, University of Strathclyde, Glasgow, Scotland, G1 1XH

*Correspondence to*: Robert J. Wilson (rwi@pml.ac.uk)

**Abstract.** Data on Secchi disk-depth (the depth at which a standard white disk lowered into the water just becomes invisible to a surface observer) show that water clarity in the North Sea declined during the 20[th] century, with likely consequences for

marine primary production. However, the causes of this trend remain unknown. Here we analyze the hypothesis that changes in the North Sea's wave climate were largely responsible, by increasing the concentrations of suspended particulate matter (SPM) in the water column through re-suspension of seabed sediments. First, we analyzed the broad-scale statistical relationships between SPM and bed shear stress due to waves and tides. We used hindcasts of wave and current data to construct a space-time dataset of bed shear stress between 1997 and 2017 across the northwest European Continental Shelf, and

compared the results with satellite-derived SPM concentrations. Bed shear stress was found to drive most of the inter-annual variation in SPM in the hydrographically mixed waters of the central and southern North Sea. We then used a long-term wave reanalysis to construct a time series of bed shear stress from 1900 to 2010. This shows that bed shear stress increased significantly across much of the shelf during this period, with increases of over 20% in the south eastern North Sea. An increase in bed shear stress of this magnitude would have resulted in a large reduction in water clarity. Wave-driven processes are rarely

included in projections of climate change impacts on marine ecosystems, but our analysis indicates that this should be reconsidered for shelf sea regions.

## 1 Introduction

The vertical attenuation of light with depth in the oceans is a complex process of scattering and absorption by particulate and

dissolved materials in the seawater. Particulate material may be biogenic or mineral. In the open ocean the majority of particles are biogenic, but this is often not the case in shallow shelf seas. Here, mineral suspended particulate matter (SPM) from land-runoff, dust deposition and re-suspension of seabed sediments can form the majority of light-scattering and absorbing material. The sub-surface light environment is a major determinant of primary production and the feeding behaviour and vertical distribution of visual predators (Dupont and Aksnes, 2012), so natural and anthropogenic factors which affect SPM

concentrations in shelf seas have the potential to cause cascading trophic effects throughout food webs (Heath et al. 2016).

Modern electronic instrumentation for measuring underwater light has only a recent history, but in many areas of the world there is a long archive of Secchi disk-depth measurements. These refer to the depth at which a standardized white disk lowered

into the sea just becomes invisible to a surface observer. Secchi disk measurements have been collected in the North Sea since the early 1900s and show long-term declines in water clarity in the central and southern North Sea (Capuzzo et al., 2015; Dupont & Aksnes, 2013). It is speculated that these changes have impacted the ecology, including a decline in North Sea primary productivity over the last 25 years (Capuzzo et al., 2017).

A shallowing of Secchi disk depth by approximately 50% over the 20th century (Capuzzo et al., 2015) needs to be explained. Secchi disk depth might be expected to depend on both SPM and phytoplankton (chlorophyll) concentrations. However, empirical data show that by far the strongest correlation is with SPM, with a roughly 1:1 relationship between percentage change in SPM, and percentage reduction in Secchi disk depth (Håkanson, 2006). Hence we might expect the reported changes in the North Sea to be associated with substantial increases in SPM.

There are insufficient empirical data extending back to the early 20$^{th}$ century to directly estimate changes in SPM in the North Sea. An alternative is to analyze physical environmental drivers that might have led to changes in SPM. These might include widespread sediment disturbance due to waves and tides and more local disturbances due to human activities (trawling, dredging and mining), coastal erosion and inputs from river discharges (Capuzzo et al, 2015). Here we examine the scope for changes in SPM to have been caused by trends in natural bed-shear stress arising from the combination of tidal and wave

climate. We can assume that the tidal regime in the North Sea has remained constant over the 20$^{th}$ century, despite there being some minor long-term changes due to long-period tides (Wunsch, 1967) and the influence of climate (De Dominicis et al., 2018). In contrast, hindcast simulations based on historical weather data consistently show that there has been a large increase in significant wave height in the North Sea (Vikebø et al., 2003; Weisse et al., 2012) and the entire northeast Atlantic (Gulev & Grigorieva, 2006).

This study has three aims. The first is to assess if there is evidence that the reductions in Secchi Disk depth occurred across the entire southern and central North Sea during the 20$^{th}$ century. The second is to establish the broad scale statistical relationship between inter-annual changes in bed shear stress and SPM. This relationship will show the potential sensitivity of SPM to long-term changes in bed shear stress. The third aim is to construct a time series of bed shear stress on the European Continental Shelf covering the years from 1900 to 2010. We then use the changes in bed shear stress to estimate the changes

in SPM caused by changes in wind regime.

## 2 Methods

2.1 Summary and overview

The region of study was the northwest European Continental Shelf, which we define to be regions shallower than 150 meters

between 48°N and 62°N and 13°W and 9°E. This region has large variation in seabed sediments (Wilson et al., 2018), wave and tidal conditions (Neill et al., 2014; Semedo et al., 2015), suspended sediment (van der Molen et al., 2016), and natural and anthropogenic disturbance of the seafloor (Aldridge et al., 2015; Diesing et al., 2013).

Our two key goals were to 1) quantify the large-scale temporal relationship between bed shear stress and SPM across this region between 1997 and 2017 and 2) quantify how bed shear stress changed during the period 1900-2010. This choice of time periods was largely a result of the availability of data (table 1). High-resolution spatiotemporal maps of SPM are available through the conversion of satellite reflectance data to SPM (e.g. Gohin et al., 2011), and these are available for 1997 onwards.

5   The calculation of bed shear stress requires data on bathymetry, sediment properties, and wave and tidal parameters. We assumed that long-term changes in bed shear stress were caused largely by changes in wave regime on the grounds that changes in tidal parameters are likely to be minimal. Bed shear stress was therefore calculated using climatological tidal conditions, bathymetry and sediment properties. Wave parameters are available from high resolution reanalysis products that cover both the present day and the 20[th] century. However, no single product exists that covers both the historical period we are interested

10   in and the entirety of the satellite era. We therefore chose to calculate bed shear stress during the satellite SPM era using the ERA-interim reanalysis, which will provide the best available relationship between bed shear stress and SPM. For the 20[th] century, we used the ERA20c reanalysis which provides high resolution wave parameters for the period 1900-2010. All analysis was carried out at a spatial resolution of 0.125° by 0.125°.

**Table 1: Data used for calculation of bed shear stress and stratification indices over the time periods 1997-2017 and 1900-2010. Calculations were calculated at a spatial resolution of 0.125° by 0.125°, and all data was available at that or finer resolution.**

| | Data | Source | Time period | Reference |
|---|---|---|---|---|
| 1 | Tidal velocities | Scottish Shelf Model | Climatology | De Dominicis et al. 2017 |
| 2 | Present day wave parameters | ERA-interim reanalysis | 1997-2017 | Dee et al., 2011 |
| 3 | Historical wave parameters | ERA20c | 1900-2010 | Poli et al., 2016 |
| 4 | Surface non-algal suspended particulate matter | Copernicus-Globcolour | 1997-2017 | Gohin 2011 |
| 5 | Surface and seabed temperature | Atlantic-European North West Shelf – Ocean Physics Reanalysis | 1997-2014 | O' Dea et al., 2012 |
| 6 | Bathymetry | General Bathymetric Chart of the Oceans | - | GEBCO |
| 7 | Sediment parameters | Gridded median grain size map | - | Wilson et al., 2018 |

**2.2 Suspended particulate matter and stratification**

Monthly and 8-day estimates of non-algal surface SPM (g m$^{-3}$) from September 1997 to August 2017 were calculated using Copernicus-Globcolour data, which is derived from satellite observations using the algorithm of Gohin (2011). We used the Copernicus Marine Environment Monitoring Service (CMEMS, http://marine.copernicus.eu) global ocean SPM reprocessed

product (QUID: http://resources.marine.copernicus.eu/documents/QUID/CMEMS-OC-QUID-009-030-032-033-037-081-082-083-085-086-098.pdf ; PUM: http://resources.marine.copernicus.eu/documents/PUM/CMEMS-OC-PUM-009-ALL.pdf). SPM data from Copernicus-Globcolour were available at 4 km resolution, and we therefore interpolated monthly SPM onto the 0.125° by 0.125° grid using bilinear interpolation.

A seasonal climatology of near-surface SPM for the years 1997 to 2017 is shown in Fig. 1. SPM shows large spatial variation. Deeper regions typically have lower SPM due to lower rates of sediment re-suspension, greater sea-surface altitude above the seabed, and remoteness from river inputs. Coastal SPM is notably elevated near river plumes such as in the East Anglian Plume and near the Severn Estuary. Furthermore, there is a notable seasonal cycle, with SPM lowest in summer.

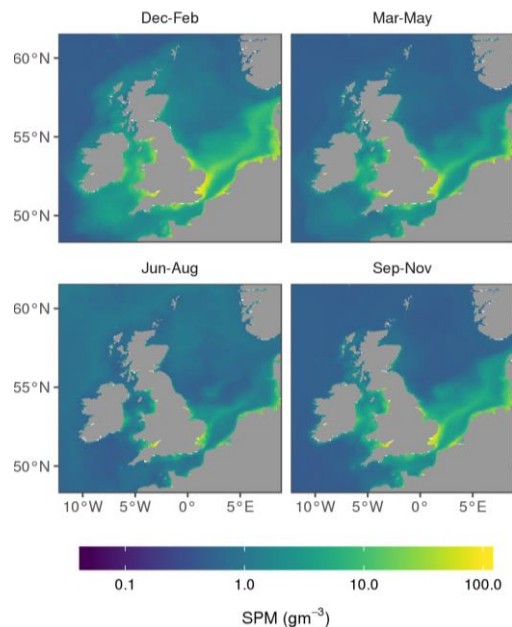

**Figure 1: Seasonal climatologies of non-algal suspended particulate matter at the water surface for the period 1997-2017. SPM was calculated from Copernicus-Globcolour estimates which are derived from remote sensing data using the Gohin (2011) algorithm.**

SPM concentrations throughout the water column are strongly determined by the combination of net re-suspension flux from the sediments and vertical mixing and turbulent diffusivity (Heath et al., 2017). Seasonal thermal stratification may therefore play a role in determining patterns of SPM in the near-surface waters. We therefore created a stratification index to analyze this influence.

Differences between surface and seabed temperature is a commonly used proxy for stratification level (Coma et al., 2009), and a difference of 0.5 °C has previously been used to reference the onset of seasonal stratification in the North Sea (Quante et al., 2016). The water column was therefore classified as being stratified when the difference between the surface and seafloor temperature was greater than 0.5 °C. Daily temperatures were taken from the MetO-NWS-REAN-PHYS-004-009 reanalysis product, available from CMEMS. This reanalysis (PUM: http://resources.marine.copernicus.eu/documents/PUM/CMEMS-

NWS-PUM-004-009.pdf; QUID: http://resources.marine.copernicus.eu/documents/QUID/CMEMS-NWS-QUID-004-009.pdf) uses the FOAM-AMM7 operational forecast system (O'Dea et al., 2012). Daily seabed temperatures from 1997-2014

were interpolated onto each 0.125° by 0.125° grid point and we then calculated the average percentage of days the water column was stratified during each month. We then calculated a monthly climatological average for the period 1997-2014.

**2.3 Bed shear stress calculations and inputs**

Bed shear stress was calculated using an existing algorithm based on the vector combination of orbital velocities due to wave action and directed flows due to tidal and other barotropic and baroclinic currents over smooth and rough beds. The shear velocity due to tidal flow was calculated using the water column averaged velocity using the "law of the wall" method (Soulsby & Clarke 2005). This assumes that there is a logarithmic decrease in velocity with proximity to the sediment-water interface. Wave orbital velocities at the seabed were calculated using the equations of Soulsby (2006). Bed shear stress was then calculated using the equations of Soulsby & Clarke (2006), and this accounts for both the magnitude and the relative direction of tides and waves.

In most shallow shelf sea situations, the flows are dominated by tides. The physical inputs necessary for these equations are depth-averaged current speed and direction, wave orbital velocity amplitude and orientation at the seabed, bed roughness, and bathymetry. The wave orbital velocity calculations require data on significant wave height, wave direction, wave period and bathymetry as inputs. The equations used to calculate bed shear stress were summarized in Wilson et al. (2018).

Bathymetry data were acquired from the General Bathymetric Chart of the Oceans (GEBCO). Data were downloaded at 30 arc second resolution from https://www.gebco.net/data and products/gridded bathymetry data/. We then used bilinear interpolation to re-grid this to 0.125° by 0.125° resolution.

Depth-averaged tidal velocities were derived from an unstructured grid, finite-volume 3D hydrodynamic model FVCOM ("The Scottish Shelf Model"; De Dominicis et al., 2017). A one-year climatology (1990-2014) of atmospheric forcings was used to run the model. We assumed that annual changes in currents (mostly due to tides) have a negligible influence on inter-annual changes in bed shear stress.

Wave fields (significant wave height, period and direction) were acquired from reanalysis products from The European Centre for Medium-Range Weather Forecasts (ECMWF). The reanalysis product ERA-interim (Dee et al., 2011) provides a state of the art reconstruction of global climate from 1979 to the present day. This provided the wave fields necessary for bed shear stress calculations over the period 1997 to 2017.

Bed shear stress was hindcast further back in time using ERA20c (Poli et al., 2016), which provides a reanalysis of global climate from 1900 to 2010. Six hourly wave direction, period and significant wave height are available for this period. All ECMWF data was downloaded from the ECMWF website https://www.ecmwf.int at 6-hourly and 0.125° by 0.125° resolution.

Bed roughness was calculated from the median grain size maps of Wilson et al. (2018), which were developed using a combination of legacy sediment composition data and environmental conditions to create synthetic maps of median grain size across the northwest European shelf. Using the inputs outlined above, bed shear stress was calculated at 15 minute intervals over the 1900-2010 and 1997-2017 periods, across the entire 0.125° latitude x 0.125° longitude grid. Bed shear stress in wave-dominated regions will be more sensitive to changes in wave regime than in areas dominated by tides. We

therefore mapped the ratio of wave-only bed shear stress to combined wave and tide bed shear stress. This gives an approximate indication of the contribution of waves to total bed shear stress.

## 2.4 Secchi disk depth data in the 20th century

Secchi Disk depth data were acquired from ICES (https://ocean.ices.dk/Project/SECCHI/), NOAA's World Ocean Database (https://www.nodc.noaa.gov/OC5/WOD/secchi-data-format.html) and Cefas (https://www.cefas.co.uk/cefas-data-hub/dois/cefas-historic-secchi-depth-measurements/). The data include all those analyzed by Capuzzo et al. (2015), and were subject to the same screening and checking.

Changes in Secchi disk depth were mapped by identifying comparable samples from before and after 1950 (following Capuzzo
et al., 2015). First, we split the samples into spring, summer, autumn and winter periods. Then for each Secchi disk record before 1950 we compared it with the mean Secchi disk depth of all post-1950 samples within 50km.

## 2.5 Statistical modelling

The sensitivity of SPM to bed shear stress is not straightforward because biological activity causes seasonality in the
consolidation of seabed sediments which affects the resistance of particulate material to resuspension (Heath et al., 2017). Hence, we analyzed the statistical relationship between SPM and bed shear stress on a month-by-month basis across years. Linear regressions were created for each 0.125° by 0.125° grid point and for larger compartmentalised regions in the North Sea. The response variable for the regressions was each available data point for 8-day SPM and the mean bed shear stress for that time period was treated as the predictor.

Illustrative large-scale relationships were also evaluated by dividing the North Sea into 10 2° by 2° compartments (Fig. 3). For each month between September 1997 and August 2017, mean SPM and bed shear stress was calculated for each compartment. In some boxes there were months when the northern cells occasionally lacked satellite SPM coverage due to cloud cover. We therefore only used cells which had 20 years of estimates. This prevents the possibility of spurious inferences due to sampling bias. For each month and compartment, we then created linear regressions of mean SPM and bed shear stress.

All statistical analysis was carried out in the statistical environment R, and data was manipulated principally using the R package dplyr (Wickham & Francois, 2016). The bed shear stress calculations were written in C++ and implemented as an R library using the package Rcpp (Eddelbuettel et al., 2011). Figures were produced using the package ggplot2. Spatial interpolation and processing of netcdf files were carried out using the Climate Data Operators tool (Schulzweida, 2017).

## 3. Results

The mapped changes in Secchi disk depth (Fig. 2) show that Secchi disk depth declined (implying increasing SPM) across broad areas of the southern and central North Sea. However, this was potentially not uniform. Comparisons of pre- and post-1950 samples show a clear decline in Secchi disk depth in regions south of 53° N. Here, 89% of the comparisons (n-99) showed declines of Secchi disk depth, with Secchi disk depth declines of over 50% predominating, and this trend is consistent across seasons. In contrast, most of the comparisons in the region north of 53° N and west of 2° E show increases, implying declining SPM and increasing water clarity. However, the number of samples in this region is small, and this should only be viewed as indicative evidence that this region did not see declines in Secchi disk depth.

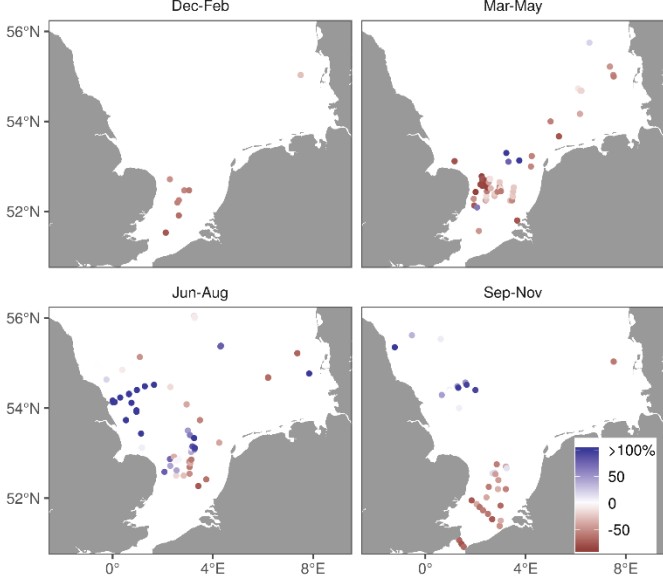

Figure 2: **: Seasonal changes in Secchi disk depth in the 20th century. Changes were estimated by detecting observations pre- and post-1950 that were in the same season and within 50 km of each other.**

Climatologies of bed shear stress and an approximation of the relative contribution of waves to bed shear stress are shown in Fig. 3. Bed shear stress is typically highest in low bathymetry coastal environments where surface energy can be more easily propagated to the seabed. However, some deep regions such as the English Channel also have high bed shear stress due to strong tidal currents. The contribution of waves to bed shear stress shows large spatial variation. Regions such as the English Channel, Norwegian Trench and Irish Sea are almost completely dominated by tidal forces. In contrast, much of the eastern North Sea and western Irish coasts are wave-dominated Bed shear stress follows a seasonal pattern because of the general

decline of wave energy between winter and summer, and this is reflected by the large reduction in the role of waves in shear stress during summer months. For example, there is a pronounced decline in the role of waves in the Northern North Sea during summer months.

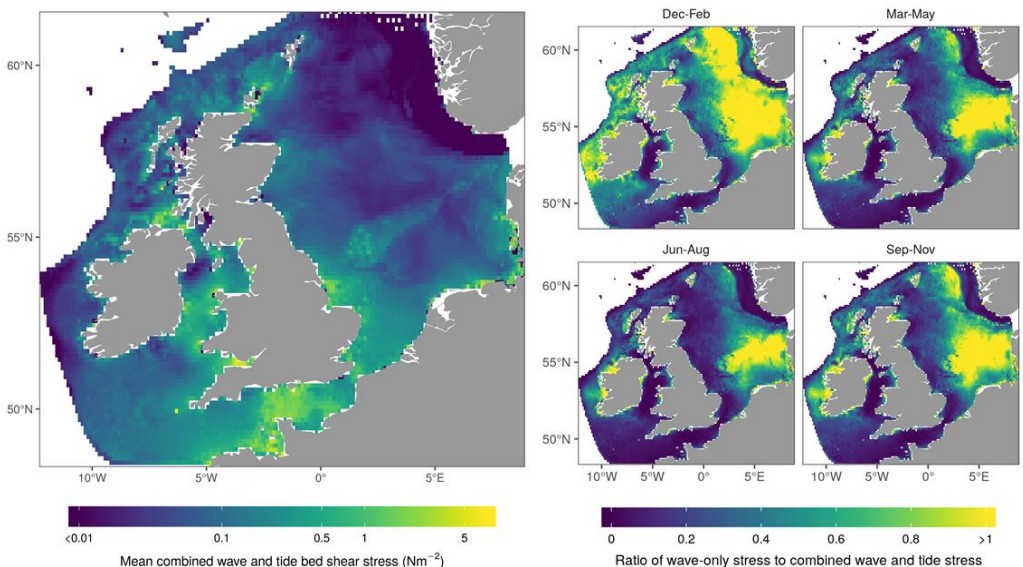

**Figure 3. Left: Annual mean of combined wave and tide bed shear stress for 1997-2017. Right: ratio of mean wave-only stress to combined wave and tide stress. Regions with high ratios are wave-dominated, whereas regions with low ratios are tide-dominated. Bed shear stress was calculated using the equations of Soulsby and Clarke (2005) and tidal inputs**
**from the Scottish Shelf Model and wave inputs from the ERA-interim reanalysis.**

Fig. 4 shows the $R^2$ value for the linear regressions of 8-day SPM and bed shear stress using data from 1997 to 2017, and stratification throughout the year. The relationship between bed shear stress and SPM shows a seasonal switch. When the water
column is vertically mixed, there is a clear positive relationship between SPM and bed shear stress across most of the study domain. The only exceptions are for river plumes such as the East Anglian Plume and the Severn Estuary, where SPM trends are likely driven by river flow and not the influence of waves on bed shear stress, and for tide dominated regions such as the English Channel where wave variation has little effect on bed shear stress. The inability of bed shear stress to explain SPM trends during stratified conditions implies that the depth and strength of the thermocline is the dominant influence on surface
SPM through its influence on the vertical profile of SPM.

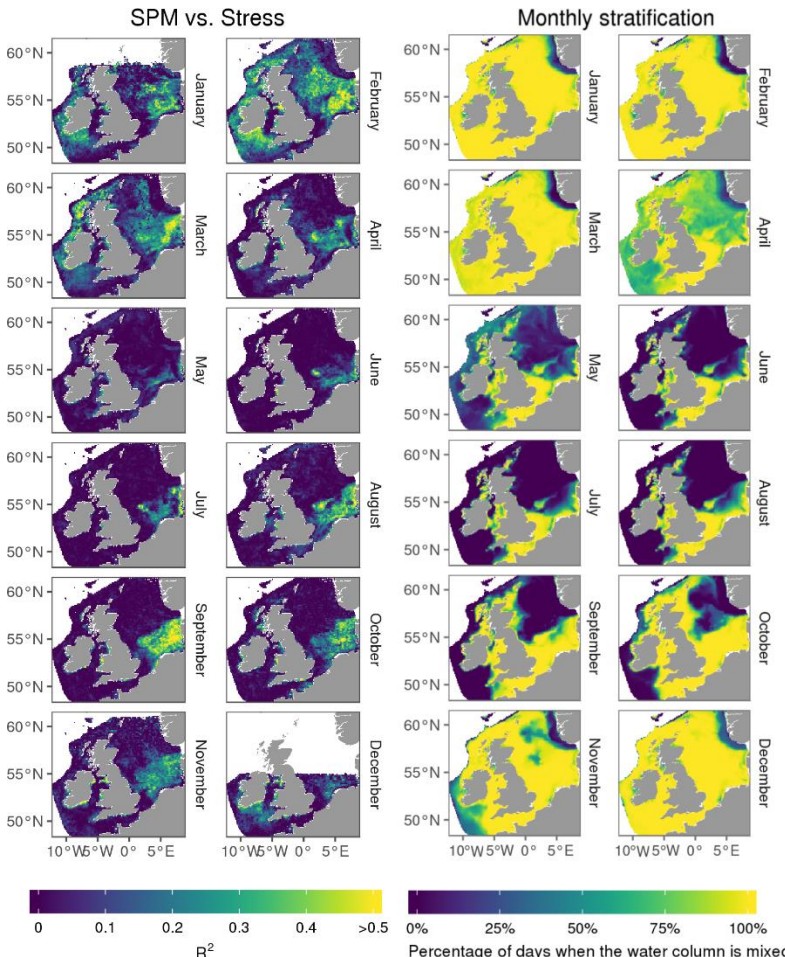

**Figure 4. Monthly relationship between bed shear stress and SPM, and stratification. The left panel shows the monthly correlation coefficient between mean monthly SPM and bed shear stress in each 0.125° by 0.125° grid point between 1997 and 2017. The right panel shows the percentage of days when the water column is mixed, with the water column classified as mixed when the difference between the sea surface and seafloor temperature is less than 0.5° C. White areas in the region are those with insufficient satellite coverage for regressions.**

In contrast, the development of a stratified water column in spring sees the decoupling of trends in SPM from bed shear stress. The transition from mixed to stratified systems in spring is reflected in a significantly reduced relationship between SPM and bed shear stress. In Autumn the spread of mixed waters is reflected by an increased positive relationship between SPM and bed shear stress, shown clearly by comparing SPM and bed shear stress south west of Ireland and in the central northern North

5  Sea in November.

Regional monthly regressions of SPM versus bed shear stress in the North Sea (Fig. 5) show a clear positive relationship between the two parameters. Comparison of the monthly regressions shows that the slope of the regression line is approximately the same across months. In contrast, the absolute values show a significant seasonal pattern. SPM declines after January once bed shear stress is controlled for. Linear regressions (Table 2) are significant ($p<0.05$) across almost all months

10  and regions, and for the majority $p<0.01$. Time series of annual SPM in these regions show that there has been an apparent decline in SPM between 1997 and 2017 (Fig. S1). This trend is consistent with the decline in mean bed shear stress over the same period (Fig. S2).

**Table 2: Results of linear regressions of monthly SPM and bed shear stress. – indicates there is insufficient data for a regression. P-values below 0.01 are indicated using \*.**

| | Parameter | 1 | 2 | 3 | 4 | 5 | 6 | 7 | 8 | 9 | 10 | 11 | 12 |
|---|---|---|---|---|---|---|---|---|---|---|---|---|---|
| *Jan* | Intercept | -0.20 | - | - | -0.84 | -3.03 | -0.99 | 1.25 | 2.40 | -0.22 | -1.81 | 5.89 | -2.1 |
| | Stress | 21.52 | - | - | 68.79 | 64.87 | 28.75 | 5.69 | 6.53 | 80.16 | 56.78 | 25.39 | 53.86 |
| | $R^2$ | 0.12 | - | - | 0.28 | 0.43 | 0.28 | 0.12 | 0.15 | 0.69 | 0.63 | 0.09 | 0.71 |
| | p-value | 0.13 | - | - | 0.02 | <0.01 | 0.02 | 0.14 | 0.10 | * | * | 0.19 | * |
| | | | | | | | | | | | | | |
| *Feb* | Intercept | 0.15 | -0.64 | -0.06 | -0.32 | -3.29 | -0.39 | -0.18 | 0.94 | -1.43 | -0.89 | -0.10 | -0.46 |
| | Stress | 11.81 | 36.90 | 19.86 | 37.30 | 86.74 | 19.00 | 12.51 | 13.14 | 95.30 | 51.45 | 50.78 | 49.96 |
| | $R^2$ | 0.046 | 0.38 | 0.48 | 0.57 | 0.53 | 0.23 | 0.26 | 0.28 | 0.75 | 0.71 | 0.37 | 0.54 |
| | p-value | 0.37 | * | * | * | * | 0.03 | 0.02 | 0.02 | * | * | * | * |
| | | | | | | | | | | | | | |
| *Mar* | Intercept | -2.14 | -0.53 | 0.04 | -0.19 | -1.52 | -1.49 | -0.35 | 0.31 | -0.51 | -0.84 | 2.61 | -0.57 |
| | Stress | 31.86 | 31.24 | 15.50 | 32.69 | 58.68 | 27.57 | 13.16 | 14.27 | 77.27 | 46.18 | 29.95 | 43.60 |
| | $R^2$ | 0.55 | 0.48 | 0.47 | 0.44 | 0.52 | 0.48 | 0.51 | 0.44 | 0.57 | 0.53 | 0.24 | 0.42 |
| | p-value | * | * | * | * | * | * | * | * | * | * | 0.02 | * |
| | | | | | | | | | | | | | |
| *Nov* | Intercept | -0.52 | 0.08 | 0.08 | -0.61 | 0.35 | -0.71 | -1.135 | -0.60 | -0.16 | 1.70 | -0.29 | 1.34 |
| | Stress | 16.96 | 11.84 | 12.56 | 33.23 | 25.41 | 18.23 | 15.88 | 19.70 | 55.28 | 19.77 | 33.76 | 25.33 |
| | $R^2$ | 0.14 | 0.31 | 0.41 | 0.56 | 0.30 | 0.26 | 0.51 | 0.57 | 0.53 | 0.27 | 0.28 | 0.26 |
| | p-value | 0.10 | 0.01 | * | * | 0.01 | 0.02 | * | * | * | 0.02 | 0.02 | 0.02 |

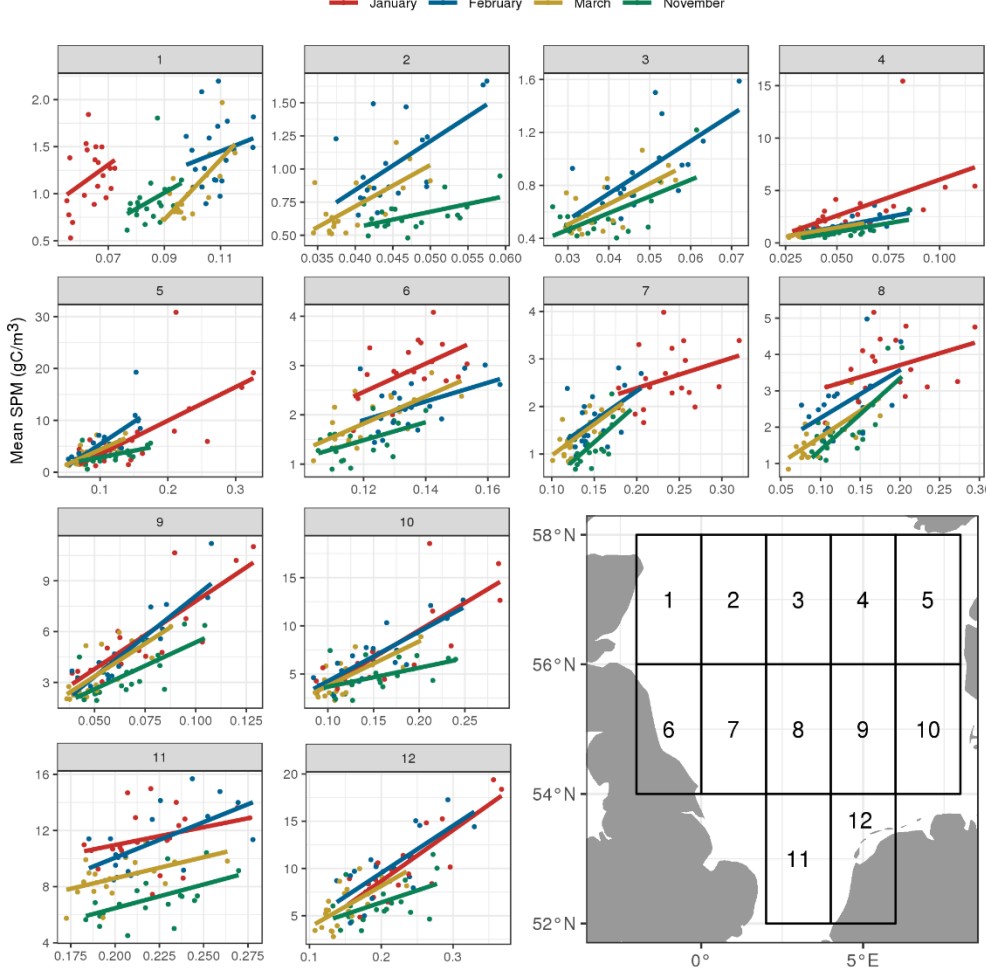

**Figure 5. Monthly regressions of mean SPM and bed shear stress for binned regions in the North Sea. The coloured lines represent regressions for the months January, February, March and November. For each mapped box and month, grid points with a 20-year record of SPM are selected. The regression of monthly SPM and bed shear stress is then derived by averaging the selected values in each. Months shown are those where the water column is permanently mixed, with December ignored due to poor data coverage.**

Significant wave height data for the 20th century and the reconstructions of bed shear stress show that there were pronounced changes over the 20th century across much of the region (Fig. 6). Bed shear stress increased significantly across the entire shelf between the periods 1910-1929 and 1990-2019. However, there was pronounced geographic variation in these changes. The largest changes occurred in the central and eastern North Sea, and the northwest of Ireland. These increases were driven

primarily by the geographic pattern of significant wave height changes, which similarly increased across the shelf over this period. In the southeast North Sea, increases of over 20% in significant wave height occurred over this time. In contrast, the northwest North Sea saw increases of only approximately 5%. The relative role of waves in determining bed shear stress (Fig. 2) has a strong influence on spatial variation in the changes. Regions such as the English Channel and Irish Sea that are tide dominated witnessed almost no changes in bed shear stress because any increase in waves will result in only a minor increase

in overall stress. This also plays a key role in the North Sea, where the strong east-west difference in the pattern of bed shear stress is strengthened by the fact that the eastern North Sea is much more wind dominated.

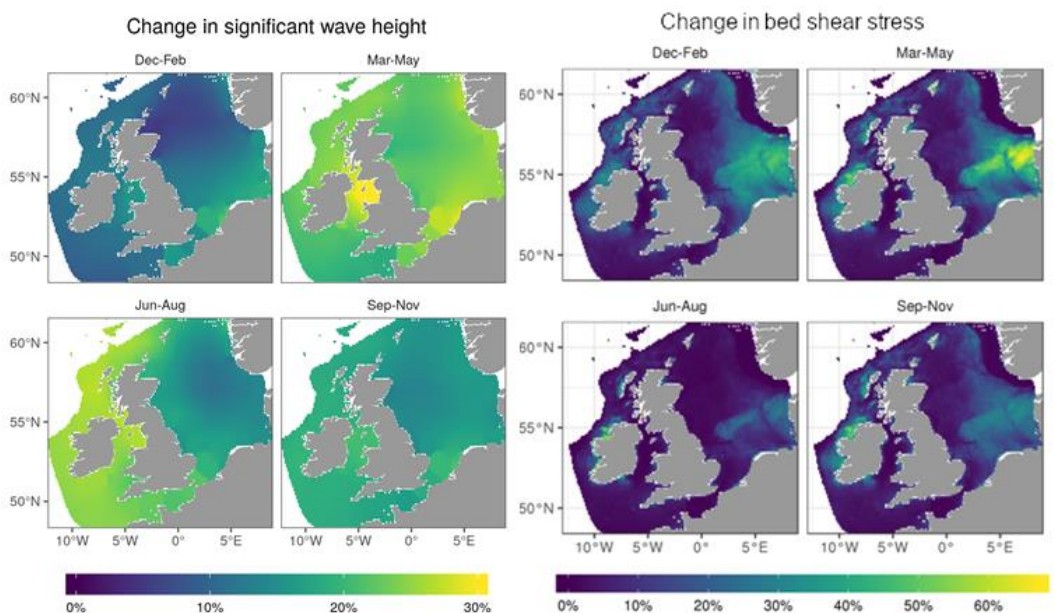

**Figure 6. Left: changes in seasonal mean significant wave height over the 20th century. Right: changes in seasonal bed**
**shear stress over the 20th century. Changes shown are between the climatological periods 1910-1929 and 1990 and 2009.**
**Bed shear stress was calculated over the 20th century using the ERA20c wave reanalysis and present day tidal**
**conditions, with significant wave height calculated using the ERA20c reanalysis wave heights.**

Estimated changes in SPM due to changes in bed shear stress over the 20th century show increases of >50% in January-March in the eastern North Sea (Fig. 4). In contrast, the western North Sea shows smaller increases of approximately 10%. There are negligible changes in tide-dominated areas such as the English Channel.

## 4. Discussion

We have shown that changes in the North Sea wave climate probably caused large reductions in water clarity during the 20[th] century. Since 1997, changes in SPM concentrations derived from remote sensing data are clearly correlated with changes in wave-induced bed shear stress, at least during winter months and in well-mixed regions of the shelf away from river plumes. Over the longer term, large parts of the southern and central North Sea experienced large increases in bed shear stress which would have had a large impact on water clarity.

Our results show that bed shear stress increased by over 20% in the eastern North Sea during the 20[th] century and rather less (around 10%) in the western and northern areas. Based on the regional regressions (Fig 4, Table 2), these changes would be expected to result in corresponding changes in SPM of approximately 10% (western) and 20% (eastern). This is corroborated to some extent by the historical Secchi disk depth data, which show that decreases in Secchi disk depth (implying increases in SPM) may have been concentrated in the southern and eastern North Sea (Fig. 4). Unfortunately, Secchi disk depth data are

sparse in the western and northern North Sea, and it is impossible to rule out that observed changes are due to random environmental effects. Improvements in the historical Secchi disk depth record are possible, but perhaps unlikely. Online text mining approaches have already been used to locate data in published literature (Aarup, 2002), and these are all incorporated into the NOAA data set used here. There is also no evidence of the existence of other accessible Secchi disk depth data for the northern North Sea, which is sparsely covered in the databases used here, despite anecdotal reports of the use of the device in

early 20[th] century oceanographic surveys.

A key motivation for this study was the recent use of trends in light attenuation (Devlin et al., 2008) to impute changes in gross primary production (Capuzzo et al. 2017) and phytoplankton phenology in the North Sea (Opdal et al., 2019). Capuzzo et al. concluded that North Sea primary production declined during the period 1988-2013, and that this trend influenced zooplankton abundances and fish recruitment, implying climate-induced 'bottom-up' changes in in the North Sea food web. Similarly,

Opdal et al. (2019) concluded that phytoplankton bloom was delayed during the 20[th] century. A key driver of these conclusions was an apparent increase in light attenuation over their study periods.

Our combined analysis of bed shear stress and satellite remote sensing data on SPM shows that the 20[th] century decline in water clarity has reversed to some extent since 1997 (Fig. S1 and S2). Between 1997 and 2017, SPM and bed shear stress showed a declining linear trend in almost all parts of the North Sea (implying decreasing light attenuation). This is in direct

contrast with the conclusions of Capuzzo et al. (2015, 2017), who determined that light attenuation had an increasing trend up to 2013. Indeed, this was a major factor in their imputed decline in gross primary production. However, Capuzzo et al. (2017) based regional estimates of light attenuation on sparsely distributed field measurements of light and SPM vertical profiles from

ships and moorings rather than remote sensing. Inspection of the spatial distributions of their field data suggests a possible bias leading to greater sampling of river plumes in later years. We therefore caution that resolving the spatial variations in SPM is necessary for robust estimates of the regional light environment and estimates of primary production trends.

Regressions of satellite remote sensing SPM versus bed shear stress across the southern North Sea when waters are mixed

indicate that approximately half of the annual variance in SPM can be explained by bed shear stress. However, this is likely an underestimate of the influence of bed shear stress on SPM. 8-day estimates of SPM are uncertain due to the sporadic nature of the remote sensing data because of cloud cover. In addition, there is an imperfect relationship between satellite-derived SPM and in situ SPM, potentially because of a failure to distinguish between SPM and phytoplankton in some months (Jafar-Sidik et al. 2017). This implies that there will be a significantly large error term in our linear regressions due to differences

between satellite and in situ SPM, and that the $R^2$ values reported here under-estimate the influence of bed shear stress on SPM.

Variations in biological activity in seabed sediments may also affect the sensitivity of SPM to bed shear stress. Heath et al. (2017) showed that seasonal patterns of in situ turbidity at a study site in the north-western North Sea could not be fully explained by variations in bed shear stress alone. It was argued that seasonal consolidation of sediments due to biological

activity significantly affects re-suspension and hence turbidity. This phenomenon is well known in shallow estuaries but less so in deeper offshore regions. Here we provide further evidence for this effect. Monthly regressions of SPM versus bed shear stress showed that SPM has a seasonal component, even when bed shear stress is controlled for, and the analysis is restricted to times when the water column is mixed. Importantly, there is evidence that benthic communities have declined in parts of the North Sea (Capuzzo et al. 2015), which potentially influenced biological activity and sediment resuspension.

Geographic variations in the sensitivity of bed shear stress to wave properties will also affect the relationship with SPM. We would expect total bed shear stress to be insensitive to wave energy in areas where tidal currents are strong. Hence, tide-dominated regions, such as the English Channel, show little proportional change in bed shear stress over the 20th century.

We have shown that wave climate is an important control of water clarity and hence light attenuation. A potential continued increase in wave energy, and thus reduction in water clarity would act as a further climate change pressure on the North West

European Shelf ecosystem. However, current projections using global assessments indicate that significant wave height will decline over the 21st century (Hemer et al., 2013, Wang et al., 2014, Wang et al., 2015), with greater reductions under higher emissions scenarios (Aarnes et al., 2017). However, the picture appears to be more complex when looking at regional models, which project clear increases in significant wave height in the eastern, but not western North Sea (Grabemann et al., 2015). Projections of the future impact of anthropogenic climate change on the world's shelf sea ecosystems should therefore consider

the future evolution of wave regimes.

**Code availability**

The code used for the calculation of bed shear stress is available as an R package on GitHub (https://github.com/r4ecology/bedshear).

**Data availability**

During peer review, monthly time series of bed shear stress from 1900-2010 and from 1997 to 2017 are available in netcdf format from https://strathcloud.sharefile.eu/d-s20cbaea41204964a and https://strathcloud.sharefile.eu/d-sdc05715d3c547f18 respectively. Post peer review, data will be made available with a permanent DOI.

**Author contributions**

RJW drafted the manuscript and RJW and MRH contributed equally to its refinement. MRH was responsible for funding acquisition. Data analysis was carried out by RJW. Bed shear stress R package was created by RJW and MRH. Figures were prepared by RJW.

**Acknowledgements**

This paper received funding under the NERC Marine Ecosystem Programme (NE/L003120/1), and from the EPSRC TeraWatt and EcoWatt2050 projects (EP/J010170/1 and EP/K012851/1). We thank Michaela De Dominicis for providing outputs from the Scottish Shelf Model.

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
