# Peer review of "Increasing turbidity in the North Sea during the 20th century due to changing wave climate"

_Ocean Science, 2019_

## Referee Comment (RC1) · Anonymous Referee #1 · 12 Jun 2019

**Review of the manuscript "Increasing turbidity in the North Sea during the 20th century due to changing wave climate", by Wilson and Heath.**

**General comments**

The authors use historic and current proxies for suspended particulate matter concentration (SPM) and bed shear stress (BSS). It is their goal to show that changes in both have been related since 1900. I like the general idea of the manuscript. It has potential and can advance the knowledge on the changes in the North Sea, but it needs significant improvement at this stage, especially to facilitate the understanding of what data are used and what assumptions are made. A large amount of data from many sources is downloaded. The description in the manuscript is very difficult to follow. I strongly advice a table where the following fields are shown: URL or FTP address, short description, date range, spatial and temporal resolution.

In particular, there is an inaccuracy in the satellite SPM data description. What authors are downloading is the CMEMS global optics L4 reprocessed product, which in turn is generated from Globcolour data. Therefore, a proper reference to the CMEMS web page is expected, as well as to the product QUID and PUG. For instance, here is the QUID document of the product they are using: http://resources.marine.copernicus.eu/documents/QUID/CMEMS-OC-QUID-009-030-032-033-037-081-082-083-085-086-098.pdf

In fact, and as a side comment, I am surprised to see a paper led by a scientist from PML download a Globcolour dataset when the CCI dataset could be used instead: http://resources.marine.copernicus.eu/documents/QUID/CMEMS-OC-QUID-009-064-065-093.pdf. This product provides global marine reflectances and has been developed by PML scientists, having a much better characterization of quality and uncertainties. This product contains Rrs(670) which can safely be used as a surrogate for turbidity. Other products like particle backscattering and also newer releases can be checked outside CMEMS, in the CCI website.

See also a typo in the ftp address provided in the manuscript, "cems". Here, the authors downloaded the global product and resampled for the North Sea instead of directly downloading the product for the European North-West Shelf Seas, for the obvious reason that a REP product is not available for the for that region. Even though this is a complaint and authors did well, it must be stated in the manuscript as a need for service improvement.

The water temperature is another CMEMS product. The URL indicated needs to be specific to that product, as well as the reference to the product QUID and PUG documents.

Though not being a specialist in the physics of the ocean, I understand that a key point in the authors' is that BSS is caused by waves, which is caused by wind. Apparently all the physics is in another paper, but it would be good if authors did a summary of how one physical quantity determines another.

Section 2.4: Methods should be described in a comprehensive for any scientist independently of the software used. So I would prefer a description in terms of equation rather than mentioning R packages.

The Secchi disk analysis is a weak point of the paper that undermines the analysis of the historic trends. A rigorous trend analysis must be made and the significance of such trends must be made. Also it is likely that authors missed many samples by forcing samples before and after 1905 to be close in space. Here I advise the authors to divide the North Sea in areas to cluster the Secchi disk measurements. Then the corresponding time series would be decomposed in seasonal, long-term and irregular trends using an approach like the X11. In open areas where data is expected to be more scarce but also less horizontal variability is expected, regions would be of greater size than coastal areas.

The paper as it is now seems to effectively prove a link between SPM and BSS for the satellite era (Figs. 1-3) but I cannot say the same based on the historic period (Fig. 4). Assuming the Secchi disk analysis correct, which I am not sure about, I am unable to appreciate any relationship between the left and right panels of Fig. 4, and the related discussion in Fig. 4 seems misleading. I particularly disagree with the sentence "Over the longer term, spatial patterns in changes in bed shear stress between 1910-1929 and 1990-2009 correspond with spatial patterns in differences between Secchi disc depth pre- and post-1950."

Regressions in Fig. 3 lack their statistical parameters. How were they calculated? What weight is given to outliers? What do a similar slope but different intercept mean in terms of physics?

The analysis left as a supplement may be of interest for the main manuscript if is accordingly treated. Here, authors seem to find a reversal of the long-term water darkening, accompanied with a corresponding decrease in BSS. As commented above for the historic period, trends have to be rigorously calculated and tested for significance.

"Our analysis shows that changes in wave energy have been a key, and probably the dominant driver of changes in water clarity in the North Sea." That is a very strong statement and I would like authors to spend some time explaining the physics behind.

**Minor comments**

The page numbering restarts at every page, which I am not sure is due to journal format, but a unique numbering for the whole manuscript would help.

The words "disc" and "disk" are found in the manuscript. Authors might unify the grammar choice.

Correct "Capuzz"

The reference Jafar-Sidik et al. (2017) is not found in the reference list.

Page 6, line 5: "drive" should be "driven"

Page 2, line 6: "SMP"

Page 9, line 5: replace "are" with "is"

---

## Referee Comment (RC2) · Jochen Wollschlaeger (Referee) · 1 Jul 2019

General comments The authors relate historical observations regarding water transparency (Secchi-disk depth) with calculations of shear bed stress based on model hindcast simulations. By this, they demonstrate that increased mobilization of suspended particulate matter is a major driver for the negative trend in tranparency that was observed in the last century in the area of the North Sea. I really like the concept and the idea of the paper. Although the trend of decreasing overall transparency in the North Sea over the last century is already known (as the authors show well with their comprehensive literature review), the reasons and drivers remain largely speculative to

date. In this respect the current work contributes to understand some of the underlying processes. However, in some cases, the conclusions drawn are not always supported clear enough by the data shown. In this respect, I would recommend improvement of the manuscript. Hopefully, the remarks and questions below are helpful in this context.

Introduction: As I understand the linear regression given in Håkanson (2006), it shows a linear relationship between the log values of SPM concentration and Secchi disk depth. Thus, changes in one parameter are transferred logarithmically to the other. Therefore, I would be careful with the "20 % increase in SPM" statement (also in the discussion), inasmuch as it is based on the average decrease in Secchi disk depth Capuzzo et al. found.

Methods: Page 3, Line 4: Does diffusivity play really a role in this context? If so, please elaborate a little bit more on that and/or give a citation. Page 3, Line 7: What is the rationale behind the 0.5 °C difference as threshold for a stratified water column? If this is a common value, please refer to the appropriate literature. Page 3, Line 27+: Could you explain why you are using two different datasets for calculating bed shear stress hindcasts? Wouldn't it be better to use the larger one in terms of being consistent in the data over the whole period (although missing the years 2011 to 2017)? Page 4, Line 26: What means "Core data analysis" in this context?

Results: Page 5, Line 8-9: From my point of view, the seasonal pattern is not readily visible in Figure 1 (right). Page 6, Line 3-4: That the relation is positive is not visible from the $R^2$ values given in Figure 2. Maybe refer also to Figure 3 at this point. Furthermore, I would soften the statement "across almost the entire study domain", because even when the water column is mixed, there are some exceptions (as also stated by the authors). However, beside the two plume regions mentioned, also the English Channel, the Irish Sea, as well as the whole British east coast appear poorly impacted by the shear stress in terms of SPM. Page 7, Line 1-5: If the relation between shear bed stress and SPM is decoupled in the stratified season, what are then the drivers for the Secchi-disk decline in these months? Or is in this season also the decline in Secchidisk depth lower? If so, the authors could refer to the appropriate literature or show the respective data. Page 7, Line 10+: Maybe incorporate the change in the trend into the main manuscript, as it is interesting and contributes to the whole story. Page 9: The authors emphasize the strong decline in Secchi-disk depth south of 53°N (Figure 4, right side), and explain it with an pronounced increase in shear stress across the region. However, according to the left side of the figure, I cannot see that the decrease in Secchi-disk depth at this point correlates to an increase in bed shear stress, which appears to be relatively small in this area (approx. 0-20%). However, as in this area the East Anglian plume as well as the plume of the Rhine is present, I would rather think that the decline in Secchi-disk depth here might be controlled by changes in e.g. river outflow (as stated by the authors before). Nevertheless, for the Northeastern part of the area (53-56°N, 4-8°E) the relationship appears to be valid, although the number of data points is comprehensively small.

Discussion: Page 10, Line 13-14: I think this statement is too strong. Instead I would claim that according to the data available shear bed stress is probably an important parameter in order to explain the transparency decrease in the last century. Page 11, Line 12-18: Maybe some of the discrepancies could also be explained by a seasonally variable contribution of the organic (e.g. phytoplankton) part of SPM. Turbidity is also influenced by the presence of pelagic phytoplankton.

Minor comments Page 3, Line 14-15: Check the brackets for the reference. Page 6, Line 1: "and SPM" after bed shear stress appears to be doubled. Page 7, Line 7: Maybe replace "bed shear stress and SPM" with "the two parameters" to avoid doubling of the terms with the begin of the sentence. Page 11, Line 9 + 13: "in situ" instead of "in-situ" Caption Figure 1: In the text is stated that the bed shear stress calculations are calculated after Soulsby & Clarke (2005), but in the caption stated Soulsby (2006). Please explain or correct. Caption Figure 4: "Century" or "century"; please keep consistent

---

## Referee Comment (RC3) · Anonymous Referee #3 · 17 Jul 2019

General Comments: The authors investigate the relationship between suspended particulate matter (SPM) and bed shear stress (BSS) by means of historic, satellite and model data. They motivate well in their literature review that decreasing water clarity in the North Sea may be linked to increased SPM content. The premise of this work is enticing. It can help to motivate further research and provide an explanation for the long term increase of water turbidity. I find the paper to be well written, language wise, and the motivation and analysis part to be comprehensive, but the analysis needs to be more quantitative. Particularly, I like the message that changing wave regimes should not be neglected in long term simulations with reference to climate change. There are some details and nit-picks that need reworking.

[Figure]

Introduction: The statement of SPM increase possibly exceeding 20% needs to be made cautiously. While the method of Hakanson 2006 is perhaps not ideal to show this, I am more concerned about the vague phrasing. I find no basis for it.

The authors mention several times that tides can be assumed to be free of long-term changes, which is not exactly true, pedantically speaking. There are long period tides (see e.g. Wunsch 1967), which may be negligible directly due to their low amplitude (<1cm) but they play a role in low frequency climate oscillations. Furthermore, sea level rise has an effect on the tidal regime. It is certainly more feasible to neglect them, but then perhaps this should be mentioned.

Methods: There is a vast amount of data used and it would be helpful to expand on the particular choices of data sources and organise it for the reader's eyes (perhaps in a table or figure). Some data was taken from CMEMS/MetO-NWS-REAN-PHYS to determine if a water column was stratified (section 2.1), but depth-averaged velocities were taken from an FVCOM model, while those same velocities are available at CMEMS as well. It would be helpful to motivate the individual data choices. There may have been easier choices for a unified data set with fewer independent sources.

Page 3, line 4 (diffusivity): as a physicist, I do understand the role of diffusivity, given that turbulent diffusivity is of course in orders similar to sinking velocities. So perhaps just add the word "turbulent" there. The threshold of $0.5°C$ appears arbitrary and needs further explaining. One could e.g. refer and compare to the definition of the mixed layer depth (MLD) used in CMEMS/MetO-NWS-REAN-PHYS or Kara et al. 2000. Alternatively, one could just use said CMEMS data of the MLD instead of coming up with a new one (i.e. if the MLD is smaller than the water depth, the column is stratified). Because the ERA-interim and ERA20c are different data sets, they cover a combined period of 1990-2017. It should be made clear that there is no combined data set or otherwise how a potential integration is carried out and bias is made impossible.

I am unfamiliar with R, but as far as I can see, no tremendously complex statistical oper-
ations have been carried out that would require elaboration beyond textbook knowledge and I would know how to achieve the same results in MATLAB. However, it is perhaps helpful to provide some algorithms as flow diagrams in a supplement.

Results: The results section starts off with an explanation of a seasonal climatology of near surface SPM, as well as BSS. This would be more suitable in the methods chapter. Figure 1 is the first of several cases where the authors say in the text that there was something to see in the figure which is actually hard to see (in this case the seasonal cycle of BSS, which is noted in the text but not well visible in the image).

For figure 2, the same criticism applies as for figure 1: the text says that there is a clear positive relationship, but the figure shows dark blue colours on the left panels in several areas where the right panels show bright yellow. It needs to be made clearer what constitutes as a "clear positive relationship", i.e. a by threshold value or something of the sort and the colour maps need to be modified accordingly. For example, in the text (page 7, line 4ff) it says that the transition to mixed water increases the link between SPM and BSS, which can be seen south west of Ireland in November, but in the figure, it is dark blue there, which indicates a weak correlation. The message that figure 2 carries could be made clearer also by an area correlation, which is more quantitative than a visual comparison.

The monthly stratification was not previously described as climatological, as it is presented here in figure 2. Instead it was written in section 2.1 that the percentage of stratified water columns was taken for each month over a period of 20y. Since the analysis covers 20y and the BSS is assumed to change due to changes in wind stress, the stratification would potentially show trends as well due to changed turbulent mixing. Climatological stratification thus makes less sense than monthly means over 20y, unless it can be shown or motivated that the change in stratification is negligible. In the first paragraph of page 7, it says that SPM and BSS become uncoupled in stratified regions in summer months. However, with reference to figure 1, the authors claim a seasonality in both parameters. What is the reason for the uncoupling? Can it be

explained?

The description of the methodology for figure 3 belongs in the methods chapter (perhaps 2.4), not the figure caption, and it needs elaboration. It says "For each mapped box and month, grid points with a 20y record of SPM are selected." Are all grid points within a box selected for which there are 20y of continuous data, or are these random choices? Were the grid points that do not have continuous data coverage neglected? Why was a complete area average unfeasible? Furthermore, the authors again say that there is a "clear positive relationship", which is easy to see e.g. for box 12, but not so much for e.g. box 1. Regression parameters are in the supplemented tables, but it would be much handier if they were besides the respective plots as well (at least R2).

Also, all boxes are of the same size, but some are only partially covered with water, some cross widely different domains, physically speaking (e.g. box 11 covering parts of the Rhine and East Anglia plume, but reaching close to the Dogger Bank, box 5 covering the Norwegian trench and thus depths from 50-300m). This may be as a minor nit-pick, but it could be argued that a more appropriate choice of boxes could have been made (e.g. as in Capuzzo et al. 2015 or O'Driscall 2014/ICES boxes). In figure 4, the changes in Secchi depth are marked as blue and red, yet ranging from +50 to +50 to positive. Red is presumably negative, i.e. a decline, so it should be -50% there. I really struggle with the sentence p.9, l.5. The evidence indicates an increase, more than it indicates a decline or no change (in that the plotted points are blue, and strongly so). Unless the data coverage is sufficient to make a claim, a claim should not be made. Perhaps a measure of certainty should be given (e.g. through marker size).

Again for figure 4, the relationship between decline in Secchi depth and bed stress change is hard to see at first. This may be due to sparsity of data, and as the authors stated earlier, the SPM content in the areas south of 53°N are heavily influenced by river intrusions. Hence, modifying the map by highlighting areas of high river intrusions could help clarify the link between the left panels and the right. Furthermore, a less selective method of data comparison than choosing points with 50km of each other

might help here as well.

At p.9, l.8, it says that there was a significant increase in BSS across the entire shelf between 1910-1929 and 1990-2019. This is immensely confusing, because previously (figure S2), a trend of decreasing BSS was shown for the latter period, so it invokes the understanding that the two periods are investigated individually, and not against each other. Perhaps this could be clarified by rephrasing. In the same paragraph it says that the changes are driven by increased significant wave height (SWH), which could be shown in a figure, e.g. by an area correlation. Is there literature as to why the increases in SWH were so variant over space?

Discussion: In the first paragraph of the discussion, it is argued that there is a decline in primary production (PP), which is attributed to reduced clarity. However, figure S1 shows decreasing trends in SPM. There is a need for elaboration as to why there can be declining SPM as a main contributor to turbidity and reduced PP. The authors later provide this elaboration on page 11, but for easier understanding, the two paragraphs should be interwoven. As a side note: a large number of Secchi depth measurements are taken from near shore stations, e.g. the NIOZ facility on Texel, NL. This will heavily skew measurements in a surrounding area.

The statement in line 13-14 is too strong and needs to either be more strongly motivated (quantitatively), or weakened. In line 19 on page 10, it is again said that there would be an expected increase of 20% in SPM. There needs to be a source for this claim and a direct reference as to how this claim can be made.

In lines 12ff on page 11, biological activity is mentioned as a potential impact on SPM and BSS. Note that before 1950, larger areas of the North Sea had benthic flora (see e.g. Capuzzo et al. 2015), which impacts BSS heavily (and thus also tides, as to my earlier point).

Minor comments: A search for typos and grammar mistakes is appropriate. My main points of criticism are with the figures, as stated above.

Page 6, line 1: "Fig. 2 shows the R2 value for the linear regressions of 8-day SPM and bed shear stress and SPM for each month between 1997 and 2017, and stratification throughout the year." This sentence is hard to follow. Perhaps it should be "... the linear regressions of 8-day SPM and bed shear stress [...] for each month..."? Line 2: "The relationship between bed shear stress shows a seasonal switch.", seems to be missing that extra "SPM" from line 2. Line 5: "... driven... ".

---

## Author Comment (AC4) · 4 Sep 2019

We again thank all of the reviewers for their helpful comments. The responses helped us clarify certain weaknesses in the text, especially in the figures. We have therefore modified two of the figures to provide greater clarity about the model results.

The first figure shows 1) an annual climatology of bed shear stress and b) the ratio of wave-only to combined wave and tide bed shear stress. The first version of the manuscript showed a seasonal climatology of bed shear stress. However, we agree with the reviewers that the seasonal pattern is not easily visible. We have therefore switched to a simpler annual climatology. The reviewers' comments made us realize

that a key part of the story was missing from the original manuscript. The long-term influence of waves on bed shear stress is not simply down to how much waves change, but whether a region is wave or tide dominated. For example, in tide dominated areas, changes in waves will make little difference. We have therefore added a panel showing a climatology of the ratio between wave-only bed shear stress and combined wave and tide bed shear stress. This shows that regions such as the English Channel will experience little change due to waves, while in the North Sea there is a strong east-west difference in the relative influence of waves. This helps explain some of the geographic patterns in 20th century shear stress.

The second modified figure combines the 20th century changes in bed shear stress with that of significant wave height. Again, this was prompted by the comments of the reviewers. This shows, in combination with the previously mentioned figure, that the geographic pattern in changes in bed shear stress are due to both changes in waves and whether how wave-dominated a region is.

The figures are attached as a supplementary pdf.

Please also note the supplement to this comment:
https://www.ocean-sci-discuss.net/os-2019-52/os-2019-52-AC4-supplement.pdf

**Supplement:**

[Figure]

**Figure. Left: Annual mean of combined wave and tide bed shear stress for 1997-2017. Right: ratio of mean wave-only stress to combined wave and tide stress. Regions with high ratios are wave-dominated, whereas regions with low ratios are tide-dominated. Bed shear stress was calculated using the equations of Soulsby and Clarke (2005) and tidal inputs from the Scottish Shelf Model and wave inputs from the ERA-interim reanalysis.**

10

[Figure]

**Figure. Left: changes in seasonal mean significant wave height over the 20[th] century. Right: changes in seasonal bed shear stress over the 20[th] century. Changes shown are between the climatological periods 1910-1929 and 1990 and 2009. Bed shear stress was calculated over the 20[th] century using the ERA20c wave reanalysis and present day tidal conditions, with significant wave height calculated using the ERA20c reanalysis wave heights.**

---

## Author Response (AR1)

**Responses to: Review of the manuscript "Increasing turbidity in the North Sea during the 20th century due to changing wave climate", by Wilson and Heath.**

**General comments**

The authors use historic and current proxies for suspended particulate matter concentration (SPM) and bed shear stress (BSS). It is their goal to show that changes in both have been related since 1900. I like the general idea of the manuscript. It has potential and can advance the knowledge on the changes in the North Sea, but it needs significant improvement at this stage, especially to facilitate the understanding of what data are used and what assumptions are made. A large amount of data from many sources is downloaded. The description in the manuscript is very difficult to follow. I strongly advice a table where the following fields are shown: URL or FTP address, short description, date range, spatial and temporal resolution.

In particular, there is an inaccuracy in the satellite SPM data description. What authors are downloading is the CMEMS global optics L4 reprocessed product, which in turn is generated from Globcolour data. Therefore, a proper reference to the CMEMS web page is expected, as well as to the product QUID and PUG. For instance, here is the QUID document of the product they are using: http://resources.marine.copernicus.eu/documents/QUID/CMEMS-OC-QUID-009-030-032-033-037-081-082-083-085-086-098.pdf

A data table has now been added, and Globcolour references have been corrected. We have clarified the use of CMEMS data by providing the approprotitate links to the relevant product user manuals.

In fact, and as a side comment, I am surprised to see a paper led by a scientist from PML download a Globcolour dataset when the CCI dataset could be used instead: http://resources.marine.copernicus.eu/documents/QUID/CMEMS-OC-QUID-009-064-065-093.pdf. This product provides global marine reflectances and has been developed by PML scientists, having a much better characterization of quality and uncertainties. This product contains Rrs(670) which can safely be used as a surrogate for turbidity. Other products like particle backscattering and also newer releases can be checked outside CMEMS, in the CCI website.

The practical reason for this is that the lead author was working at the University of Strathclyde while this paper was written. The original submission erroneously did not provide a double affiliation, but this has now been fixed in the resubmission.

The use of Rrs_670 reflectance was something we considered during the analysis. However, we expect that the paper is targeted at a very broad audience, in particular marine ecologists, and we believe that using marine reflectance, and not SPM will reduce the accessibility of the paper. Furthermore, while there are some benefits to using the OCCCI data products, it is unclear to us if they are superior for our purposes. The critical issue is that we use a product that best captures temporal changes in turbidity. As far as we know there has yet to be an empirical comparison of various products in this regard.

See also a typo in the ftp address provided in the manuscript, "cems". Here, the authors downloaded the global product and resampled for the North Sea instead of directly downloading the product for the European North-West Shelf Seas, for the obvious reason that a REP product is not available for the for that region. Even though this is a complaint and authors did well, it must be stated in the manuscript as a need for service improvement.

We have now removed references to the ftp sites. CMEMS often change where data is stored, so adding an ftp that is likely to become redundant is probably a bad idea.

We are unsure that the paper is an appropriate venue for calls for service improvements for CEMS. If we do this for CEMS, we should do this for the other data providers, who sometimes have a greater need than CEMS to improve services.

The water temperature is another CMEMS product. The URL indicated needs to be specific to that product, as well as the reference to the product QUID and PUG documents.

We have now added relevant QUID and PUB documents.

Though not being a specialist in the physics of the ocean, I understand that a key point in the authors' is that BSS is caused by waves, which is caused by wind. Apparently all the physics is in another paper, but it would be good if authors did a summary of how one physical quantity determines another.

Section 2.4: Methods should be described in a comprehensive for any scientist independently of the software used. So I would prefer a description in terms of equation rather than mentioning R packages.

All of the equations used in the analysis are supplied in Wilson et al. 2018. We have now made this clearer. We believe that is more appropriate to point readers there instead of repeating the equations in new supplementary material.

The Secchi disk analysis is a weak point of the paper that undermines the analysis of the historic trends. A rigorous trend analysis must be made and the significance of such trends must be made. Also it is likely that authors missed many samples by forcing samples before and after 1905 to be close in space. Here I advise the authors to divide the North Sea in areas to cluster the Secchi disk measurements. Then the corresponding time series would be decomposed in seasonal, long-term and irregular trends using an approach like the X11. In open areas where data is expected to be more scarce but also less horizontal variability is expected, regions would be of greater size than coastal areas.

We have now moved the Secchi disk analysis to a separate figure, which shows changes by season. The aim of this analysis was to give a prelimenary, and admittedly indicative, view of the spatial changes in Secchi Disk depth. The approach taken runs into the problem that in many regions there is sparse data, and the langugage used in the original paper was perhaps not cautious enough. However, we are skeptical that alternative approaches are available. The aggregation methods of Dupont and Aksnes (2013) and Capuzzo et al. (2015) do not effectively deal with spatial sampling bias.   For example,  Dupont and Aksnes (2015) control for bottom depth and distance to the coast. However, bed shear stress and sediment properties are likely  to be more important covariates.  Given that there  are huge differences in the spatial coverage in Secchi disk samples pre- and post-1950 we are concerned that the aggregation methods  used in other studies do not result in partly spurious trends.

The paper as it is now seems to effectively prove a link between SPM and BSS for the satellite era (Figs. 1-3) but I cannot say the same based on the historic period (Fig. 4). Assuming the Secchi disk analysis correct, which I am not sure about, I am unable to appreciate any relationship between the left and right panels of Fig. 4, and the related discussion in Fig. 4 seems misleading. I particularly disagree with the sentence "Over the longer term, spatial patterns in changes in bed shear stress between 1910-1929 and 1990-2009 correspond with spatial patterns in differences between Secchi disc depth pre- and post-1950."

We agree the original paper overstates the ability of the reanalysis to recreate the historical trends in the original figure 4. In reality, because of the sparseness of the historical Secchi Disk depth data we cannot say with huge confidence what the spatial patterns were. However, while the big picture – North Sea water clarity declined – remains in in place, there are some indications this was not universal across the North Sea. The paper's text has now been reworded to be more cautious.

Regressions in Fig. 3 lack their statistical parameters. How were they calculated? What weight is given to outliers? What do a similar slope but different intercept mean in terms of physics?

In the original manuscipt the statistical parameters were given in the supplementary materials. These have now been moved to the main text.

The analysis left as a supplement may be of interest for the main manuscript if is accordingly treated. Here, authors seem to find a reversal of the long-term water darkening, accompanied with a corresponding decrease in BSS. As commented above for the historic period, trends have to be rigorously calculated and tested for significance.

"Our analysis shows that changes in wave energy have been a key, and probably the dominant driver of changes in water clarity in the North Sea." That is a very strong statement and I would like authors to spend some time explaining the physics behind.

This statement in the original text was probably too strong. We have now reworded it to make it clear that we have shown that changes in the wave regime caused large increases in bed shear stress. Because of the spatial sparsity of the Secchi Disk depth data, there is a lot of uncertaintity in the actual changes in Secchi Disk depth, so we should have been more cautious in interpreting the results.

The key result of the paper is that the wave changes shown by the wave reanalysis would have resulted in large reductions in water clarity. The original placement of this result side by side with the Secchi disk depth data result probably provided a misleading impression. The new figures arrangement is more reflective of the key results in the paper.

**Minor comments**

The page numbering restarts at every page, which I am not sure is due to journal format, but a unique numbering for the whole manuscript would help.

Page numbering restarts in each page as part of the Copernicus default template.

The words "disc" and "disk" are found in the manuscript. Authors might unify the grammar choice.

This has now been changed to "disk" throughout.

Correct "Capuzz"

This has now been corrected.

The reference Jafar-Sidik et al. (2017) is not found in the reference list.

This has now been added.

Page 6, line 5: "drive" should be "driven"

This has been corrected.

Page 2, line 6: "SMP"

This has been corrected to "SPM".

Page 9, line 5: replace "are" with "is"

This has been corrected.

Replies to: Interactive comment on "Increasing turbidity in the North Sea during the 20th century due to changing wave climate" by Robert J. Wilson and Michael R. Heath

Jochen Wollschlaeger (Referee) jochen.wollschlaeger@hzg.de

General comments The authors relate historical observations regarding water transparency (Secchi-disk depth) with calculations of shear bed stress based on model hindcast simulations. By this, they demonstrate that increased mobilization of sus- pended particulate matter is a major driver for the negative trend in tranparency that was observed in the last century in the area of the North Sea. I really like the concept and the idea of the paper. Although the trend of decreasing overall transparency in the North Sea over the last century is already known (as the authors show well with their comprehensive literature review), the reasons and drivers remain largely speculative to date. In this respect the current work contributes to understand some of the underlying processes. However, in some cases, the conclusions drawn are not always supported clear enough by the data shown. In this respect, I would recommend improvement of the manuscript. Hopefully, the remarks and questions below are helpful in this context.

Introduction: As I understand the linear regression given in Håkanson (2006), it shows a linear relationship between the log values of SPM concentration and Secchi disk depth. Thus, changes in one parameter are transferred logarithmically to the other. Therefore, I would be careful with the "20 % increase in SPM" statement (also in the discussion), inasmuch as it is based on the average decrease in Secchi disk depth Capuzzo et al. found.

We have removed the 20% reference in the introduction, as it wasn't really necessary. The 20% increase in the discussion was referring to the approximate increase in SPM we would expect in the south eastern North Sea. This was based on the regressions for that region. The original text did not make that clear, so we have reworded it.

Methods: Page 3, Line 4: Does diffusivity play really a role in this context? If so, please elaborate a little bit more on that and/or give a citation.

The influence of diffusivity on the vertical profile of SPM is discussed in Heath et al. 2017.  We have therefore moved the reference to Heath et al. 2017 to the end of this sentence, where it is more appropriate.

Page 3, Line 7: What is the rationale behind the 0.5 ◦C difference as threshold for a stratified water column? If this is a common value, please refer to the appropriate literature.

The use of 0.5 C was motivated by the North Sea Region Climate Change Assessment, which used this as the metric for the onset of seasonal stratification. The text has been modified to make this clear, and to reference the North Sea Region Climate Change Assessment.

Page 3, Line 27+: Could you explain why you are using two different datasets for calculating bed shear stress hindcasts? Wouldn't it be better to use the larger one in terms of being consistent in the data over the whole period (although missing the years 2011 to 2017)?

Ideally, we would use one data set, not two. However, the comparison with satellite SPM was carried out to provide the best available quantitification of the large-scale influence of waves on SPM. The ERA-20c reanalysis is much less appropriate than the ERA-interim for two key reasons. First, it's temporal coverage is limited to before 2011. Second, ERA-interim is a higher quality data product.

The methods section has now been adjusted to make clearer why two separate wave reanalysis were used.

Page 4, Line 26: What means "Core data analysis" in this context?

This has been reworded to state that we were referring to the R package used for the bulk of the data manipulation.

Results: Page 5, Line 8-9: From my point of view, the seasonal pattern is not readily visible in Figure 1 (right).

After looking at this figure again, we agree the seasonal pattern is not particularly clear. We have experimented with a number of different colour scalings and have concluded that this cannot be made readily visible. Instead we have switched to just showing annual mean bed shear stress.

Page 6, Line 3-4: That the relation is positive is not visible from the R2 values given in Figure 2. Maybe refer also to Figure 3 at this point. Further- more, I would soften the statement "across almost the entire study domain", because even when the water column is mixed, there are some exceptions (as also stated by the authors). However, beside the two plume regions mentioned, also the English Channel, the Irish Sea, as well as the whole British east coast appear poorly impacted by the shear stress in terms of SPM.

The original paragraph was poorly worded. It has now been amended to make it clear that in tide-dominated regions the $R^2$ values is low.

Page 7, Line 1-5: If the relation between shear bed stress and SPM is decoupled in the stratified season, what are then the drivers for the Secchi-disk decline in these months? Or is in this season also the decline in Secchi-disk depth lower? If so, the authors could refer to the appropriate literature or show the respective data.

"Decoupled" is perhaps not a totally accurate term. What we mean to say is that when the water column is stratified variations in vertical current shear and diffusivity appear to have a much greater influence on temporal variations in surface SPM. However, this does not mean that bed shear stress does not explain the decline in water clarity during spring and summer during the 20th century. If stratification levels remained the same then bed shear stress likely drove a large part of the decline. However, whether this is true is an open question.

The results section now has a sentence stating that during stratified conditions the influence of the thermocline etc. dominates the vertical profile of SPM.

Page 7, Line 10+: Maybe incorporate the change in the trend into the main manuscript, as it is interesting and contributes to the whole story.

This an interesting part of the story, but we have reluctantly chosen to keep it in the supporting materials. The key focus of the paper is on what happened during the 20th century. Moving the two supporting figures to the main text risks undermining that, as we would have 6 figures on present day conditions, but only one on historical changes.

Page 9: The authors emphasize the strong decline in Secchi-disk depth south of 53◦N (Figure 4, right side), and explain it with an pronounced increase in shear stress across the region. However, according to the left side of the figure, I cannot see that the decrease in Secchi-disk depth at this point correlates to an increase in bed shear stress, which appears to be relatively small in this area (approx. 0-20%). However, as in this area the East Anglian plume as well as the plume of the Rhine

is present, I would rather think that the decline in Secchi-disk depth here might be controlled by changes in e.g. river outflow (as stated by the authors before). Nevertheless, for the Northeastern part of the area (53-56◦N, 4-8◦E) the relationship appears to be valid, although the number of data points is comprehensively small.

We have now moved the historical Secchi disk depth data to a separate figure. This was originally placed beside the bed shear stress figure to reduce the figure count more than anything. However, with hindsight this was likely not a good choice. Because the data is very sparse, we can only get an indicative idea of what the spatial patterns of Secchi disk depth changes were. The key issue is whether the big picture stories agree, and they largely do.

Discussion: Page 10, Line 13-14: I think this statement is too strong. Instead I would claim that according to the data available shear bed stress is probably an important parameter in order to explain the transparency decrease in the last century.

The text has been changed to be something more cautious. We have now changed the text to say large reductions in water clarity would have resulted from the bed shear stress changes shown.

Page 11, Line 12-18: Maybe some of the discrepancies could also be explained by a seasonally variable contribution of the organic (e.g. phytoplankton) part of SPM. Turbidity is also influenced by the presence of pelagic phytoplankton.

This should have been stated on page 11 lines 8-13, which referenced Jafar-Sidik who found that satellite SPM potentially mixes up SPM and phyotoplankton during summer months. The text has been amended accordingly.

Minor comments

Page 3, Line 14-15: Check the brackets for the reference.

This has been corrected.

Page 6, Line 1: "and SPM" after bed shear stress appears to be doubled.

This has been corrected.

Page 7, Line 7: Maybe replace "bed shear stress and SPM" with "the two parameters" to avoid doubling of the terms with the begin of the sentence.

Agreed. This has been changed.

Page 11, Line 9 + 13: "in situ" instead of "in-situ"

This has been corrected.

Caption Figure 1: In the text is stated that the bed shear stress calculations are calculated after Soulsby & Clarke (2005), but in the caption stated Soulsby (2006). Please explain or correct.

It should have been Soulsby and Clarke in the caption. Now corrected.

Caption Figure 4: "Century" or "century"; please keep consistent

This has now been made consistent throughout the text

Reply to: Interactive comment on "Increasing turbidity in the North Sea during the 20th century due to changing wave climate" by Robert J. Wilson and Michael R. Heath

Anonymous Referee #3

General Comments: The authors investigate the relationship between suspended par- ticulate matter (SPM) and bed shear stress (BSS) by means of historic, satellite and model data. They motivate well in their literature review that decreasing water clarity in the North Sea may be linked to increased SPM content. The premise of this work is enticing. It can help to motivate further research and provide an explanation for the long term increase of water turbidity. I find the paper to be well written, language wise, and the motivation and analysis part to be comprehensive, but the analysis needs to be more quantitative. Particularly, I like the message that changing wave regimes should not be neglected in long term simulations with reference to climate change. There are some details and nit-picks that need reworking.

Introduction: The statement of SPM increase possibly exceeding 20% needs to be made cautiously. While the method of Hakanson 2006 is perhaps not ideal to show this, I am more concerned about the vague phrasing. I find no basis for it.

While we believe the original phrasing was cautious in that 20% is within the bounds shown by some empirical studies, we have now removed the 20% reference, as it was not really necessary.

The authors mention several times that tides can be assumed to be free of long-term changes, which is not exactly true, pedantically speaking. There are long period tides (see e.g. Wunsch 1967), which may be negligible directly due to their low amplitude (<1cm) but they play a role in low frequency climate oscillations. Furthermore, sea level rise has an effect on the tidal regime. It is certainly more feasible to neglect them, but then perhaps this should be mentioned.

We have now added a reference to Wunsch, and a second reference showing the potential impacts of climate change.

Methods: There is a vast amount of data used and it would be helpful to expand on the particular choices of data sources and organise it for the reader's eyes (perhaps in a table or figure). Some data was taken from CMEMS/MetO-NWS-REAN-PHYS to determine if a water column was stratified (section 2.1), but depth-averaged veloc- ities were taken from an FVCOM model, while those same velocities are available at CMEMS as well. It would be helpful to motivate the individual data choices. There may have been easier choices for a unified data set with fewer independent sources.

A table has now been added to make it easier to understand the data used. With hindsight some easier choices could be made, however data choices were in some cases the result of what was available through certain projects., and we decided against simplifying them, given data processing methods were already in place.

Page 3, line 4 (diffusivity): as a physicist, I do understand the role of diffusivity, given that turbulent diffusivity is of course in orders similar to sinking velocities. So perhaps just add the word "turbulent" there.

The word "turbulent" has now been added to clarify the text.

The threshold of $0.5{\circ}C$ appears arbitrary and needs further explaining. One could e.g. refer and compare to the definition of the mixed layer depth (MLD) used in CMEMS/MetO-NWS-REAN-PHYS

or Kara et al. 2000. Alterna- tively, one could just use said CMEMS data of the MLD instead of coming up with a new one (i.e. if the MLD is smaller than the water depth, the column is stratified).

The choice of 0.5 C came from the North Sea Region Climate Change Assessment 2016, who defined stratified waters as those with prolonged periods with temperature differences between surface and seabed above 0.5 C. We believe this is a reasonable proxy for stratification for the purposes of this paper. The aim is to illustrate which months have mixed water columns, not to provide precise quantifications.

Because the ERA-interim and ERA20c are different data sets, they cover a combined period of 1990-2017. It should be made clear that there is no combined data set or otherwise how a potential integration is carried out and bias is made impossible.

The text in the methods section has now made this clearer, and we have stated that two reanalysis were used because the ERA20c reanalysis does not overlap fully with the satellite SPM.

I am unfamiliar with R, but as far as I can see, no tremendously complex statistical operations have been carried out that would require elaboration beyond textbook knowledge and I would know how to achieve the same results in MATLAB. However, it is perhaps helpful to provide some algorithms as flow diagrams in a supplement.

All of the equations used to calculate shear stress are provided in the supplementary materials of Wilson et al., 2018. We have now made this clear in the text. The actual bed shear stress code is written in C++, not R. This has also now been made clearer in the text.

Results:  The results section starts off with an explanation of a seasonal climatology     of near surface SPM, as well as BSS. This would be more suitable in the methods chapter.

We have now moved the SPM climatology to the methodology, with the BSS climatology still in the results. We have kept the BSS climatology in the results because we have now added a panel showing the relative contribution of waves to bed shear stress.

Figure 1 is the first of several cases where the authors say in the text that there was something to see in the figure which is actually hard to see (in this case the seasonal cycle of BSS, which is noted in the text but not well visible in the image).

We agree that this does not read very well. Because of the very large spatial variation in BSS, we have concluded that it is not possible to map seasonal BSS with the seasonal variations being particularly clear. And so we have just gone with an annual mean. The map of bed shear stress is big picture context for the main results, and is one that has appeared in various other papers. So it makes sense to stick with an annual climatology.

For figure 2, the same criticism applies as for figure 1: the text says that there is a clear positive relationship, but the figure shows dark blue colours on the left panels in several areas where the right panels show bright yellow. It needs to be made clearer what constitutes as a "clear positive relationship", i.e. a by threshold value or something of the sort and the colour maps need to be modified accordingly. For example, in the text (page 7, line 4ff) it says that the transition to mixed water increases the link between SPM and BSS, which can be seen south west of Ireland in November, but in the figure,  it is dark blue there, which indicates a weak correlation. The message that figure 2 carries could be made clearer also by an area correlation, which is more quantitative than a visual comparison.

The original text was poorly explained. We forgot to state that in tidally dominated regions waves will have little influence, so we do not expect the model to explain much. This is illustrated by the new figure 3, which approximates the relative influence of waves. The region South West of Ireland has a relatively low influence of waves. We therefore expect relatively low $R^2$ values in the regressions. It is notable that the $R^2$ values are a lot higher in this region in December and January than in November, which is attributable to higher waves.

The monthly stratification was not previously described as climatological, as it is pre- sented here in figure 2. Instead it was written in section 2.1 that the percentage of stratified water columns was taken for each month over a period of 20y. Since the analysis covers 20y and the BSS is assumed to change due to changes in wind stress, the stratification would potentially show trends as well due to changed turbulent mix- ing. Climatological stratification thus makes less sense than monthly means over 20y, unless it can be shown or motivated that the change in stratification is negligible. In the first paragraph of page 7, it says that SPM and BSS become uncoupled in strati- fied regions in summer months. However, with reference to figure 1, the authors claim a seasonality in both parameters. What is the reason for the uncoupling? Can it be explained?

The text in section 2.1 has now been clarified to make it clear we calculated a climatology.

It is true that stratification potentially changed during this time period and also during the 20th century. Analysing this was out of the scope of this paper. However, it is important to quantify this relationship. It is true that there will be complex relationships between bed shear stress and stratification due to the influence of winds. So this ought to be resolved by future work.

In general, the goal of our study was to quantify the influence of bed shear stress on sediment in the water column, not necessarily its vertical profile. This was why the linear regressions in figure 3 were carried out when waters were mixed. Moving towards accounting for stratification is something we will consider in future.

The reason for the decoupling is that variations in the depth and strength of the thermocline etc. appear to be the dominant influence on sediment reaching the surface. This is also potentially complicated by the influence of winds on both waves and stratification.

The description of the methodology for figure 3 belongs in the methods chapter (per- haps 2.4), not the figure caption, and it needs elaboration. It says "For each mapped box and month, grid points with a 20y record of SPM are selected." Are all grid points within a box selected for which there are 20y of continuous data, or are these random choices? Were the grid points that do not have continuous data coverage neglected?

We only chose grid points with a 20 year record to improve the quality of the data being used in the regressions. The points neglected are essentially those at the northern fringe of satellite coverage. These points have low reliability and we were retiscent to use then. This was an attempt to reduce the amount of variation in the data caused by poor satellite coverage. This potentially could have been developed further. It is noticeable that the R2 of the regression models are notably lower in January than in March.

A sentence has now been added to section 2.4 to clarify this.

Why was a complete area average unfeasible?

Area averaging is problematic because for some regions it can result in temporal comparisons not being apples to apples. Roughly speaking, we are trying to estimate what would happen if, say, bed

shear stress doubled in a region. However, to do this the points of comparison must be consistent. The relationship between bed shear stress and SPM varies significantly in space due to variations in sediments and bathymetry. As a result, a strict area averaging will increase the amount of noise due to poor satellite coverage and could lead to sampling bias.

Furthermore, the authors again say that there is a "clear positive relationship", which is easy to see e.g. for box 12, but not so much for e.g. box 1. Regression parameters are in the supplemented tables, but it would be much handier if they were besides the respective plots as well (at least R2).

We believe that the regression results should be interpreted carefully. A major challenge is understanding how much of the variation in SPM is caused by noise within the satellite SPM. This is particularly true for box 1, i.e. the north eastern North Sea. Our regressions show that there is a high p-value for the regressions in January, February and November, but not in March. This is potentially simply down to low light and cloud coverage creating a very noise satellite SPM record that cannot be explained in any detail by bed shear stress. Arguably some of the regressions should be removed because of this, but we believe it is better to caveat that there can be large uncertainties in the satellite SPM.

The parameters have now been moved to the main text.

Also, all boxes are of the same size, but some are only partially covered with water, some cross widely different domains, physically speaking (e.g. box 11 covering parts of the Rhine and East Anglia plume, but reaching close to the Dogger Bank, box 5 covering the Norwegian trench and thus depths from 50-300m). This may be as a minor nit-pick, but it could be argued that a more appropriate choice of boxes could have been made (e.g. as in Capuzzo et al. 2015 or O'Driscall 2014/ICES boxes).

The boxes were chosen as they could provide an estimate of the spatial variation. As with all choices they have limitations. We agree that alternatives are possible, but was unclear to us that they do not run into similar problems. For example, many of the zones used by Capuzzo et al. often cover a very broad range of shear stress, sediment and bathymetric regimes. Furthermore, our choice of boxes was motivated partly by the desire to choose regions that would have highly correlated wave climates, which makes aggregating bed shear stress over a region reasonable.

In figure 4, the changes in Secchi depth are marked as blue and red, yet ranging from +50 to +50 to positive. Red is presumably negative, i.e. a decline, so it should be -50% there.

Something went wrong when the figure was being tweaked prior to submission. We have now fixed this figure so that the legend is correct.

I really struggle with the sentence p.9, l.5. The evidence indicates an increase, more than it indicates a decline or no change (in that the plotted points are blue, and strongly so). Unless the data coverage is sufficient to make a claim, a claim should not be made. Perhaps a measure of certainty should be given (e.g. through marker size).

We believe that the original paragraph was suitably caveated, and we were clear that these results were indicative. Our key message here was that while the data show that there seems to have been a decline in water clarity across the North Sea, we cannot be sure it was universal. This was the implication of the work of Capuzzo et al. (2015), but there really isn't sufficient data to be confident of the exact spatial details of the changes.

Again for figure 4, the relationship between decline in Secchi depth and bed stress change is hard to see at first. This may be due to sparsity of data, and as the authors stated earlier, the SPM content in the areas south of 53◦N are heavily influenced by river intrusions. Hence, modifying the map by highlighting areas of high river intrusions could help clarify the link between the left panels and the right. Furthermore, a less selective method of data comparison than choosing points with 50km of each other might help here as well.

At p.9, l.8, it says that there was a significant increase in BSS across the entire shelf between 1910-1929 and 1990-2019. This is immensely confusing, because previously (figure S2), a trend of decreasing BSS was shown for the latter period, so it invokes the understanding that the two periods are investigated individually, and not against each other. Perhaps this could be clarified by rephrasing. In the same paragraph it says that the changes are driven by increased significant wave height (SWH), which could be shown in a figure, e.g. by an area correlation. Is there literature as to why the increases in SWH were so variant over space?

The clarifying phrase "between the periods" has been added to the text.

We have amended the text to make it clear that long-term spatial variation in changes in bed shear stress are not simply down to changes in significant wave height, but that the relative importance of waves in determining bed shear stress is also critical. Figure 2 now shows the ratio of wave-only bed shear stress to combined wave and tide bed shear stress. This gives an indication of the regions of tide and wave dominance. Say we have a region where 10% of stress comes from waves, and one where 50% comes from waves. If we simplify the physics and ignore interactions a doubling of wave stress in one region will result in a 10% increase in overall stress, but it would result in a 50% increase in the other. So a lot of the spatial pattern in the 20th century changes really comes down to how relatively influential waves are.

Discussion: In the first paragraph of the discussion, it is argued that there is a decline in primary production (PP), which is attributed to reduced clarity. However, figure S1 shows decreasing trends in SPM. There is a need for elaboration as to why there can be declining SPM as a main contributor to turbidity and reduced PP. The authors later provide this elaboration on page 11, but for easier understanding, the two paragraphs should be interwoven. As a side note: a large number of Secchi depth measurements are taken from near shore stations, e.g. the NIOZ facility on Texel, NL. This will heavily skew measurements in a surrounding area.

We agree. The discussion has now been modified to combine these paragraphs. We further agree with the reviewer that there is great potential for Secchi depth measurements to be skewed in the way mentioned. While some existing studies fail to account for this, we also recognize that it is often very difficult to do so.

The statement in line 13-14 is too strong and needs to either be more strongly moti- vated (quantitatively), or weakened. In line 19 on page 10, it is again said that there would be an expected increase of 20% in SPM. There needs to be a source for this claim and a direct reference as to how this claim can be made.

We have reworded the sentence to say that wave regime was a driver of the decline in water clarity.

In lines 12ff on page 11, biological activity is mentioned as a potential impact on SPM and BSS. Note that before 1950, larger areas of the North Sea had benthic flora (see e.g. Capuzzo et al. 2015), which impacts BSS heavily (and thus also tides, as to my earlier point).

Relevant text has now been added.

Minor comments: A search for typos and grammar mistakes is appropriate. My main points of criticism are with the figures, as stated above.

Page 6, line 1: "Fig. 2 shows the R2 value for the linear regressions of 8-day SPM and bed shear stress and SPM for each month between 1997 and 2017, and stratification throughout the year." This sentence is hard to follow. Perhaps it should be ". . . the linear regressions of 8-day SPM and bed shear stress [. . .] for each month. . ."?

This sentence has been tidied up

Line 2: "The relationship between bed shear stress shows a seasonal switch.", seems to be missing that extra "SPM" from line 2. Line 5: ". . . driven. . . ".

A correction has been added to the text.

[revised manuscript text omitted]